# Ex Vivo Analysis of Cell Differentiation, Oxidative Stress, Inflammation, and DNA Damage on Cutaneous Field Cancerization

**DOI:** 10.3390/ijms25115775

**Published:** 2024-05-26

**Authors:** Lara Camillo, Elisa Zavattaro, Federica Veronese, Laura Cristina Gironi, Ottavio Cremona, Paola Savoia

**Affiliations:** 1Department of Health Sciences, University of Eastern Piedmont, Via Paolo Solaroli 17, 28100 Novara, Italy; lara.camillo@uniupo.it (L.C.); paola.savoia@med.uniupo.it (P.S.); 2AOU Maggiore della Carità di Novara, c.so Mazzini 18, 28100 Novara, Italy; federica.veronese@med.uniupo.it (F.V.); gironi.laura@gmail.com (L.C.G.); 3IRCCS San Raffaele Scientific Institute, Via Olgettina 58, 20132 Milan, Italy; ottavio.cremona@hsr.it; 4San Raffaele Scientific Institute, Vita-Salute San Raffaele University, Via Olgettina 58, 20132 Milan, Italy

**Keywords:** skin cancer, field cancerization, ultraviolet light, DNA damage, oxidative stress

## Abstract

Cutaneous field cancerization (CFC) refers to a skin region containing mutated cells’ clones, predominantly arising from chronic exposure to ultraviolet radiation (UVR), which exhibits an elevated risk of developing precancerous and neoplastic lesions. Despite extensive research, many molecular aspects of CFC still need to be better understood. In this study, we conducted ex vivo assessment of cell differentiation, oxidative stress, inflammation, and DNA damage in CFC samples. We collected perilesional skin from 41 patients with skin cancer and non-photoexposed skin from 25 healthy control individuals. These biopsies were either paraffin-embedded for indirect immunofluorescence and immunohistochemistry stain or processed for proteins and mRNA extraction from the epidermidis. Our findings indicate a downregulation of p53 expression and an upregulation of Ki67 and p16 in CFC tissues. Additionally, there were alterations in keratinocyte differentiation markers, disrupted cell differentiation, increased expression of iNOS and proinflammatory cytokines IL-6 and IL-8, along with evidence of oxidative DNA damage. Collectively, our results suggest that despite its outwardly normal appearance, CFC tissue shows early signs of DNA damage, an active inflammatory state, oxidative stress, abnormal cell proliferation and differentiation.

## 1. Introduction

The concept of field cancerization (FC), first proposed by Slaughter in 1953, delineates an area surrounding a neoplastic lesion that appears morphologically normal yet harbors cellular clones exhibiting phenotypic alterations [1,2,3]. This concept is currently applied across various tissues throughout the body, including the skin [1]. Specifically, cutaneous field cancerization (CFC) predominantly affects photoexposed areas and is characterized by an elevated risk of multiple actinic keratoses (AKs) and cutaneous squamous cell carcinomas (cSCCs) [4,5,6]. However, the precise understanding of CFC remains ambiguous and subject to ongoing debate in the scientific community, especially regarding its definition, diagnostic criteria, and therapeutic strategies [7,8]. The main extrinsic factor implicated in CFC development is chronic exposure to ultraviolet radiation (UVR), which was also considered in the 8th AJCC classification for non-melanoma skin tumors [9]. UVR is known to induce driver mutations in specific target genes on some precursor cells. With prolonged UV exposure, these cells may accumulate additional mutations leading to uncontrolled cell growth and the establishment of fields of altered cells that gradually supplant the normal epithelium [3,10]. Mechanistically, UVR can cause DNA damage directly through direct absorption of UV photons by nucleotides or indirectly through reactive oxygen species (ROS) production [11,12,13], which in turn can induce oxidative lesions such as 8-hydroxy-2′-deoxyguanosine (8-OHdG) within the DNA [14,15]. Furthermore, UVR can also initiate an inflammatory response and upregulate the expression of the inducible nitric oxide synthase (iNOS) [16,17,18], collectively fostering an environment conductive to carcinogenesis. [1]. Genetic analyses reveal that the most frequently mutated genes in both normal sun-exposed skin and AKs are TP53 [6,19,20], NOTCH1-2 [21,22], and p16 [10,21,23]. Furthermore, Chitsazzadeh et al. [24] discovered mutations in NOTCH3 and FGFR3 in normal skin, while Miola et al. [25] have recently demonstrated alterations in Ki67 and surviving expression is affected in CFC. These genetic mutations highlight the profound clinical and therapeutic consequences of CFC in oncodermatology; yet, extant therapies directed at AKs and SCCs do not target the mutated cellular clones within CFC, consequently failing to prevent the recurrence of primary tumors as well as the genesis of de novo lesions [5,24]. Consequently, patients undergo multiple and invasive treatments, causing morbidities and generating tremendous costs [26]. Comprehensive understanding of all mechanisms involved in CFC formation and progression will contribute to the design of field-directed therapies to reduce and prevent AK development and the recurrence of cancer [27].

In this context, we have conducted an investigation into the expression of keratinocyte differentiation markers, pro- and anti-inflammatory cytokines, and markers of DNA damage in skin biopsies from both normal and CFC tissue.

## 2. Results

### 2.1. Expression of p53 and Ki67 Is Altered in CFC

In response to UV radiation, epidermal cells can arrest the cell cycle through the activation of p53 in order to allow DNA repair or promote apoptosis when severe mutations are present [16]. However, as demonstrated in both AKs and cSCCs, in chronic photodamaged areas p53 is often mutated, driving premature cell cycle progression and aberrant proliferation of cells before DNA damage has been repaired [28]. For this reason, p53 and Ki67, markers of cell proliferation, could be considered as indicators of field cancerization [29,30]. We therefore investigated whether the expression of normal p53 and Ki67 were altered in the CFC skin samples (Figure 1). Immunohistochemical analysis (Figure 1a) revealed that in healthy skin, both p53 and Ki67 are expressed mainly by keratinocytes of the basal layer showing a cytoplasmatic stain. On the other hand, in CFC skin we observed a significant reduction of p53 positive cells (Figure 1b) that was correlated with downregulation of the *TP53* gene (Figure 1c) and protein expression (Figure 1d,e). We also observed that in CFC tissue, in comparison with healthy skin (CTRL), Ki67 is overexpressed at both the protein (Figure 1a) and gene (Figure 1c) level.

### 2.2. Cell Cycle Is Impaired in CFC

To correctly repair UV-induced DNA damage, cells activate the expression of cyclin-dependent kinase (CDK) inhibitors, together with p53, to arrest the cell cycle [31]. To verify whether this fundamental pathway could be affected by UVR, we investigated the expression of the CDK inhibitors and tumor suppressors p16 and p21 (Figure 2). We found that both *p16* and *p21* gene expression was upregulated in CFC skin samples, although a significant difference was observed only for *p16* (Figure 2a). Through IF we observed that p16 was mostly expressed on keratinocytes of the basal layers, while p21 expression was diffused on spinous and granular layers of the epidermis (Figure 2b) of both normal and CFC samples. In accordance with gene expression results, we found that the number of cells expressing both proteins was higher in the CFC skin in comparison with normal skin, despite a significant difference was detected only with p16 (Figure 2c).

### 2.3. Analysis of Keratinocytes Differentiation Markers

CK14 and CK10 are intermediate filaments expressed by keratinocytes of the basal and spinous layers, respectively [32], while Filaggrin is expressed by full-differentiated keratinocytes on the stratum corneum [33]. Despite their important role in keratinocyte differentiation, no data are available about their expression on CFC. Hence, we evaluated their protein and gene expression in both normal and CFC biopsies (Figure 3).

Through IHC we observed that in control samples keratinocytes differentiated normally, as evidenced by the coordinate expression of CK14, CK10, and Filaggrin on their respective layers (Figure 3a), while in CFC skin, the cell differentiation process was altered. Indeed, we found that CK14 and CK10 expressions, respectively, were up- and down-regulated, evaluating both the number of IHC-positive cells (Figure 3b) and protein expression through Western blot analysis (Figure 3d,e). Analyzing the gene expression (Figure 3f), we found that CK14 and Filaggrin were down- and up-regulated in the CFC tissue, respectively, while no significant difference was found when analyzing *CK10* expression. Moreover, we recorded a slight, but not significant, increased expression of Filaggrin both in terms of IHC optical density (Figure 3c) and Western blot analysis (Figure 3d,e).

### 2.4. Oxidative Stress and Inflammation in CFC

UVR exposure triggers the generation of oxidative stress and inflammation, both contributors to DNA damage and photocarcinogenesis through the production of ROS [34]. Here, we evaluated the expression of antioxidant superoxide dismutase 1 (SOD1), iNOS, and pro- and anti-inflammatory cytokines (Figure 4). As shown in Figure 4a, we found that *iNOS* gene expression was significantly increased in the CFC samples compared with control skin, while no significant difference was observed in SOD1 expression. Similar results were observed through protein expression analysis. Indeed, no significant changes were recorded in SOD1 protein expression (Figure 4b,c), in contrast with iNOS that was expressed by a significantly greater percentage of cells (Figure 3b,d) recording high level of protein expression (Figure 4e,f). Moreover, we observed that whereas in normal skin iNOS is mainly expressed in the upper layers of the epidermis, in CFC, this protein is present also in the spinous and the basal layer.

Finally, analyzing inflammatory cytokine expression, we observed in photoexposed skin samples a remarkable overexpression of IL-6 and IL-8 and a downregulation of IL-10 at the gene level (Figure 4g).

### 2.5. Analysis of Oxidative DNA Damage

ROS can react with nucleotides inducing the formation of oxidative base lesions like the 8-OHdG that is promptly excised by the 8-oxoguanine DNA glycosylase (OGG1) enzyme [35]. If left uncorrected, this DNA damage contributes to the accumulation of gene mutations within DNA molecules [36]. Thus, we investigated the expression of 8-OHdG and OGG1 in CFC as early signs of photodamage (Figure 5). Through IHC, we observed that 8-OHdG was mostly present in the basal and spinous layers of CFC epidermis (Figure 5a), recording a higher number of positive cells compared with control skin (Figure 5b). Consequently, we found an increased *OGG1* gene expression (Figure 5c,e) in the CFC samples, which also correlated with a protein overexpression (Figure 5f) in terms of the number of positive cells (Figure 5d) and protein expression (Figure 5f,g).

## 3. Discussion

CFC is primarily responsible for initiating skin cancer, operating through a complex multistep mutagenesis process [36,37]. Ultraviolet radiation (UVR) critically affects CFC’s development and progression into neoplastic states by inducing mutations in essential growth-regulating genes, including *TP53*, *NOTCH1-2*, *p21*, and *p16* [22,24,37]. These mutations disrupt normal cellular proliferation and facilitate escape from cell cycle control mechanisms [1]. Moreover, UV exposure amplifies the production of reactive oxygen species (ROS) through several pathways, notably those mediated by nuclear factor (erythroid-derived 2)-like 2 (NRF2) [38,39], NF-kB, and p63 [40,41]. Repeated UV exposure leads to additional cellular changes, causing cells to expand and displace normal tissue, ultimately contributing to the formation of AKs and cSCCs [42]. Identifying reliable biomarkers for treatment response in CFC remains a significant challenge.

Our study unveils previously unreported insights into CFC cell and molecular biology, showing that keratinocytes at this site exhibit early signs of UVR-induced alterations, such as reduced p53 expression, increased cell proliferation, altered cell cycles, and changes in keratinocyte differentiation, alongside the presence of oxidative stress, inflammation, and DNA damage. P53, one of the most frequently mutated genes in human cancers [4,43], appears to play a critical role in the development of CFC. Multiple studies have documented the presence of p53-mutated cell clones within chronically photoexposed skin and in both AKs and cSCCs [44,45]. However, data regarding the expression of wild-type (WT) p53 in CFC were lacking. Our investigation into both gene and protein levels in our CFC samples reveals a significant downregulation compared to healthy skin, where p53 is abundantly expressed in basal layer keratinocytes. This aligns with findings by Gupta and Ramani, who observed a complete absence of p53 staining in histological specimens of oral squamous cell carcinoma (OSCC) [46]. The reduced expression of p53 might result from negative interactions with MDM2 and MDM4 [47] or from mutations leading to a truncated p53 form undetectable by monoclonal antibodies. These observations suggest that WT-p53 expression could serve as an indicator of UVR damage and potentially as a biomarker for CFC, pending further verification.

Considering p53’s role in cell cycle arrest, we also examined the expression of cell cycle inhibitors p16 and p21, which play essential roles in cell cycle control by coordinating internal and external signals and impeding proliferation at several key checkpoints (reviewed in 10.1016/j.bcp.2023.115739). We observed an upregulation of both factors, although the increase in p21 was not statistically significant compared to healthy controls, suggesting an alteration of cell cycle checkpoints in photoexposed skin. This observation is consistent with Hodges et al. [28] and Marinescu et al. [48], who have linked p16 and p21 expression with the early stages of skin tumorigenesis and with UVR-induced cell cycle arrest phases G1 and G2 [49].

Additionally, we evaluated Ki67 expression as a cell proliferation marker and found it upregulated in CFC, particularly in the basal layer where keratinocyte stem cells reside. This finding is consistent with studies by Birajdar et al. [50] and Montebugnoli et al. [51], who reported abnormal Ki67 expression in and around OSCC lesions. To our knowledge, no other studies have assessed Ki67 expression specifically in CFC.

Moreover, we investigated the expression of keratinocyte differentiation markers CK14, CK10, and Filaggrin in CFC samples. Overexpression of CK14 and Filaggrin and downregulation of CK10 were observed, although gene and protein expressions were not always concordant, possibly due to the low number of patients and their heterogeneity or to alterations in the translation process or epigenetic effects. These preliminary findings are unique as no prior studies have explored these markers in CFC. Notably, they correlate with results by Choi et al. [52] and Sun et al. [53], who have documented changes in these protein levels during AK to SCC progression and in SCCs, respectively.

Finally, we analyzed oxidative stress and inflammation markers, discovering upregulated expression of iNOS and SOD1 and altered levels of cytokines IL-6, IL-8, and IL-10 in photoexposed samples [36]. Although the impact of oxidative stress and inflammation on skin cancer has been extensively documented [54], no studies have specifically examined SOD1 and iNOS expression in the CFC. Lastly, we assessed markers of oxidative DNA damage, 8-OHdG and OGG1, finding elevated levels in CFC keratinocytes, which corroborates findings by Yoshifuku et al. [55] regarding their overexpression in AKs and SCCs.

## 4. Materials and Methods

### 4.1. Patient Population and Skin Biopsies

In this study, we enrolled between 2019 and 2023 66 subjects (48 males, 18 females; average age 72.6 ± 12.7 years) of which 41 patients (33 males, 8 females; average age 79.0 ± 9.5 years) were affected by precancerous skin lesions, non-melanoma skin cancer (NMSC), and/or cutaneous melanoma developed on cutaneous field cancerization (CFC) areas and 25 unaffected subjects were used as healthy control (CTRL) (15 males, 10 females; average age 71.2 ± 15.6 years). Patients and healthy controls were not matched; however, differences in terms of gender composition and median age were not statistically significant. The diagnosis and surgical treatment were performed at the Dermatology Unit of AOU Maggiore della Carità, Novara (Italy), and all donors signed the informed consent approved by the competent ethical board. All patients were treated according to the best clinical practice. From each subject enrolled, we collected a small biopsy at a distance not exceeding 2 cm from resection margins of perilesional skin in the case of CFC patients, and non-photoexposed areas from the control population. Age, biological gender, anatomical site of samples, and type of skin cancer diagnosed (only for CFC patients) are presented in Table 1 for CTRL subjects and Table 2 for CFC patients. Full-thickness skin biopsies (7 CTRL, 10 CFC) were formalin-fixed, paraffin-embedded (FFPE) to perform immunohistochemistry (IHC) and indirect immunofluorescence (IF), while 18 CTRL skin biopsies and 31 CFC skin biopsies were treated to extract protein and RNA from the epidermis.

### 4.2. Separation and Lysis of Epidermis

Skin biopsies were washed three times with ethanol 75% and NaCl 0.9% solution and cut into small pieces. Then, samples were incubated with Dispase II 2 mg/mL (Merck KGaA, Darmstadt, Germany) prepared in Dulbecco’s modified eagle medium (DMEM) added with penicillin/streptomycin, at 4 °C overnight. Skin biopsies were incubated for at least 1 h at 37 °C, 5% CO_2_, and the epidermis was separated, weighed, and stored on ice. Samples were lysed with TRIzol reagent (Fisher Molecular Biology, Trevose, PA, USA) for RNA extraction and with RIPA lysis buffer (Merck KGaA) supplemented with protease inhibitor cocktail (Thermo Fisher, Waltham, MA, USA) and phosphatase inhibitors for protein isolation. Tissue lysis was performed on ice using a glass homogenizer and with 1 mL of lysis buffer per 50–100 mg of tissue. 

### 4.3. Immunohistochemistry (IHC)

Four-μm FFPE skin samples were deparaffinized and rehydrated in xylene and ethanol (100%, 95%, 90% and 75%) and washed with phosphate buffer saline (PBS). Slides were heated in microwaves for 5 min at 550 W in citrate buffer pH 6.0 (Vector Laboratories, Burlingame, CA, USA) for antigen unmasking. Endogenous peroxidases and non-specific bindings were blocked by incubating with Peroxidase 1 solution (Biocare Medical, Pacheco, CA, USA) for 10 min at room temperature (RT) and with Blocking Solution Vectastain ABC system (Vector Laboratories) for 30 min at RT. Then, primary antibodies anti-p53, anti-Ki67, anti-CK14, anti-CK10, anti-Filaggrin, anti-OGG1, and anti-8-OHdG were prepared (Appendix A), and added for 1 h at RT. Subsequently, slides were incubated with secondary biotinylated antibody (Vector Laboratories) and Vectastain ABC reagent (Vector Laboratories) for 30 min at RT each. ImmPACT DAB (Vector Laboratories) was used as a chromogen. Sections were counterstained with Gill’s Hematoxylin (Bio Optica, Milan, Italy) and dehydrated by soaking with graded ethanol (75%, 95%, 100%) and xylene. Cover slips were mounted using VectaMount Mounting Medium (Vector Laboratories). Pictures were taken using an optical microscope Nikon ECLIPSE Ci and cell count was performed using ImageJ software [56] (https://imagej.net/ij/, accessed on 23 May 2024). Cell positivity was calculated as the percentage of positive cells out of the total of cells present in the full-thickness epidermis (dermis was excluded). For each patient, we counted cells from four different fields. IHC optical density (OD) was performed using ImageJ software [56]. Briefly, the epidermis was countered using the polygon sections tool and the mean grey value was measured. Then, the OD was calculated as Log (max intensity/mean intensity). For each sample, three pictures from different areas were taken and measured. 

### 4.4. Indirect Immunofluorescence (IF)

Next, 4 μm FFPE slides were dewaxed and heated for antigen retrieval with Tris-based unmasking solution pH 9.0 (Vector Laboratories) in a microwave for 5 min at 550 W. Slides were washed in PBS and incubated with blocking solution (BSA 5%, PBS 1x, Triton 0.1%) for 1 h at RT. Then, primary antibodies anti-p16, anti-p21, anti-SOD1, and anti-iNOS were diluted as indicated in Appendix A in dilution buffer (BSA 2%, PBS 1x, Triton 0.1%) and incubated for 2 h at RT. After rinsing in PBS 1x + 0.1% Tween20, slides were incubated with DAPI (1:500, Merck KGaA) and secondary antibodies anti-mouse Alexa Fluor-546 and anti-rabbit Alexa-Fluor-488 (1:500, Thermo Fisher) for 45 min at RT in the dark. Finally, slides were washed in PBS 1x and mounted with glycerol/PBS 1x solution (9:1). Pictures were taken with an optical fluorescent microscope Leica DS5500B (Leica Microsystems, Inc., Buffalo Grove, IL, USA). The percentage of positive cells was expressed as the number of iNOS-stained cells divided by the total number of cells stained with DAPI counted in full-thickness epidermis of four different fields for each sample (the dermis was excluded). SOD1 integrated density was measured using ImageJ software using the formula: mean gray value* area. 

### 4.5. Quantitative Real-Time PCR

The epidermis was separated and lysed in TRIzol reagent as described in Section 4.2. RNA was isolated according to manufacturer’s instructions. Briefly, samples were incubated with chloroform for phase separation and centrifuged at 12,000× *g* for 15 min at 4 °C. The aqueous phase was transferred in a fresh tube and isopropanol was added. Samples were centrifuged at 12,000× *g* for 10 min at 4 °C. Supernatants were discarded and RNA precipitates were washed with ethanol 75% prepared in DEPC (diethylpyrocarbonate) water and centrifuged at 12,000× *g* for 5 min at 4 °C. The excess ethanol was removed, and the RNA pellets were allowed to dry. Total RNA purity and concentration was quantified at the spectrophotometer (Nanodrop, Thermo Fisher) by measuring the optical density at 260 and 280 nm. Reverse transcription and cDNA synthesis was performed using a High-capacity cDNA reverse transcription kit (Applied Biosystems, Foster City, CA, USA) according to the manufacturer’s instructions. qRT-PCR was performed using SensiFast SYBR No-Rox kit (Bioline, London, UK) with forward and reverse primers (Appendix A) on the Bio-rad CFX96 Touch Real-Time PCR Detection System (Biorad, Hercules, CA, USA). GAPDH was used as the housekeeping gene for data normalization. 

### 4.6. Western Blotting

The epidermis was separated and lysed using RIPA buffer as described in Section 4.2. Proteins were quantified with BCA assay (Thermo Fisher) and denaturized with 2× Loading Buffer (Merck KGaA) at 95 °C for 5 min. 30 μg of denaturized proteins were loaded on SDS-PAGE gel at 10% acrylamide. Proteins were transferred to a PVDF membrane and incubated with primary antibodies anti-p53, anti-CK14, anti-CK10, anti-Filaggin, anti-OGG1, anti-iNOS, and anti-SOD1 as indicated in Appendix A. Specific secondary antibodies HRP-conjugated were added for 1 h at RT. Membranes were developed using enhanced chemiluminescence method (ECL, Biorad, Hercules, CA, USA) and acquired with ChemiDoc Imaging System (Biorad, Hercules, CA, USA). Β-actin was used for housekeeping. The relative band intensity was quantified using ImageJ software. Densitometric analysis data are expressed as protein/β-actin ratio.

### 4.7. Statistical Analysis

Statistical analysis was performed using GraphPad Prism 8 (San Diego, CA, USA). Data were expressed as mean ± SEM of *n* different samples. Statistical significance was assessed by using Student’s *t*-test among CTRL skin and CFC skin samples. Statistical significance was defined as *p* < 0.05.

## 5. Conclusions

Our study significantly advances the understanding of cutaneous field cancerization (CFC) by demonstrating, for the first time, the downregulation of wild-type p53 and an increase in cell proliferation despite high levels of CDK inhibitor expression within the CFC. Additionally, our findings reveal that chronic exposure to ultraviolet radiation (UVR) not only triggers inflammation and oxidative stress but also actively contributes to the production of reactive oxygen species (ROS) and DNA damage, thus underlining the critical molecular dynamics at play in the development of CFC. Despite the limitations of our study, such as the limited number of patients analyzed, our findings open new avenues for research on the molecular mechanisms behind CFC development, thus identifying novel targets for preventing chronic actinic damage and treating CFC. Furthermore, exploring the role of dermal changes induced by UVR in the process of field cancerization and skin cancer development could prove extremely valuable. This deeper insight could lead to significant advancements in the strategies for managing and mitigating skin cancer risks associated with chronic UVR exposure.

## Figures and Tables

**Figure 1 ijms-25-05775-f001:**
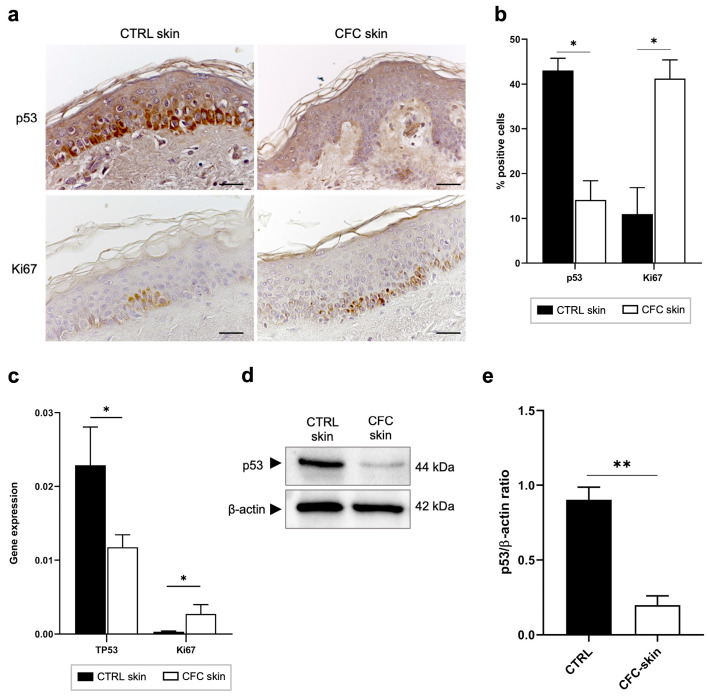
p53 and Ki67 expression is modulated on CFC skin samples. (**a**) Representative IHC and (**b**) % of p53 and Ki67 positive cells counted in four random areas of CTRL (n = 7) and CFC (n = 10) skin samples expressed as means ± SEM. (**c**) *TP53* and *Ki67* gene expression expressed as means ± SEM of CTRL (n = 12) and CFC (n = 21) skin samples. (**d**) Representative western blotting and (**e**) densitometric analysis of p53 expression tested on CTRL (n = 6) and CFC (n = 10) skin samples expressed as mean ± SEM. CTRL, control; CFC, cutaneous field cancerization; ns, no significance. Scale bar: 10 μm. * *p* < 0.05, ** *p* < 0.01.

**Figure 2 ijms-25-05775-f002:**
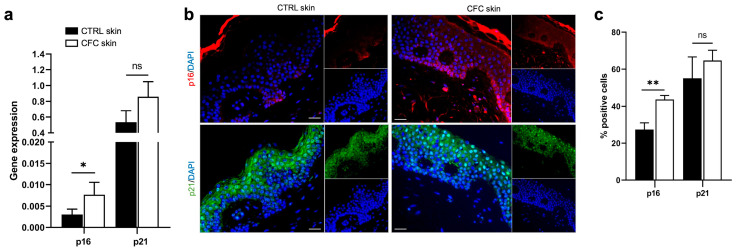
p16 expression is increased in CFC skin samples. (**a**) *p16* and *p21* gene expression indicated as means ± SEM of CTRL (*n* = 12) and CFC (*n* = 21) skin samples. (**b**) Representative IF and (**c**) % of p16 and p21 positive cells counted in four random areas of CTRL (*n* = 7) and CFC (*n* = 10) skin samples expressed as means ± SEM. CTRL, control; CFC, cutaneous field cancerization; ns, no significance. Scale bar: 10 μm. * *p* < 0.05, ** *p* < 0.01.

**Figure 3 ijms-25-05775-f003:**
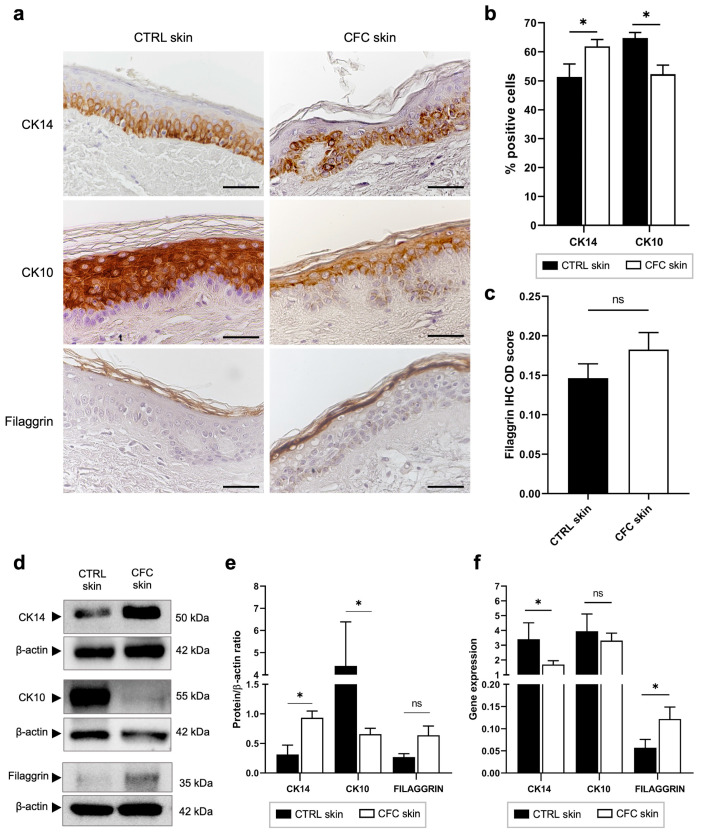
Keratinocyte differentiation is altered in CFC skin. (**a**) Representative IHC expression of CK14, CK10 and Filaggrin and (**b**) % of IHC CK14 and CK10 stain-positive cells counted in four random areas of CTRL (*n* = 7) and CFC (*n* = 10) skin samples expressed as means ± SEM. (**c**) Filaggrin IHC OD score expressed as mean ± SEM of CTRL (*n* = 7) and CFC (*n* = 10) samples. (**d**) Representative Western blot and densitometric analysis (**e**) of CK14, CK10 and Filaggrin tested on CTRL (*n* = 6) and CFC (*n* = 10) skin samples expressed as means ± SEM. (**f**) *CK14*, *CK10* and *Filaggrin* gene expression indicated as means ± SEM of CTRL (*n* = 12) and CFC (*n* = 21) skin samples. CTRL, control; CFC, cutaneous field cancerization; ns, no significance; OD, optical density. Scale bar: 10 μm. * *p* < 0.05.

**Figure 4 ijms-25-05775-f004:**
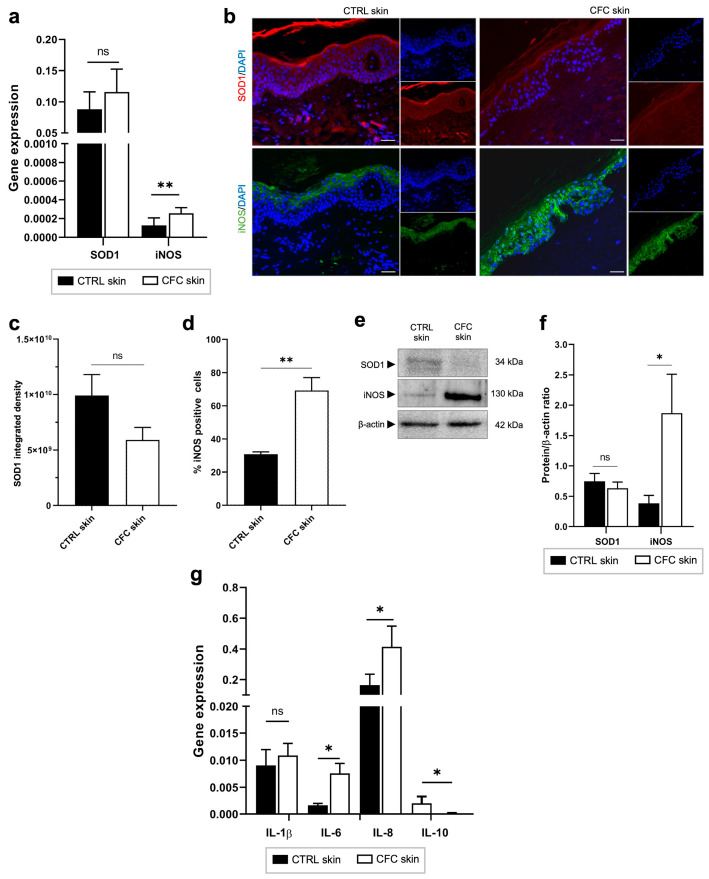
Oxidative stress and inflammatory markers are expressed in CFC skin. (**a**) *SOD1* and *iNOS* gene expression evaluated on CTRL (*n* = 12) and CFC (*n* = 20) samples and expressed as means ± SEM. (**b**) Representative IF of SOD1 and iNOS and (**c**) analysis of IF SOD1 integrated density and (**d**) % of IF iNOS positive cells expressed as means ± SEM of CTRL (*n* = 7) and CFC (*n* = 10) skin samples. (**e**) Representative Western blot and densitometric analysis (**f**) of SOD1 and iNOS on CTRL (*n* = 6) and CFC (*n* = 10) skin samples expressed as mean ± SEM. (**g**) *IL-1β*, *IL-6*, *IL-8*, *IL-10* gene expression expressed as means ± SEM of CTRL (*n* = 12) and CFC (*n* = 21) samples. CTRL, control; CFC, cutaneous field cancerization; iNOS, inducible nitric oxide synthase; SOD1, superoxide dismutase; ns, no significance. Scale bar: 10 μm. * *p* < 0.05, ** *p* < 0.01.

**Figure 5 ijms-25-05775-f005:**
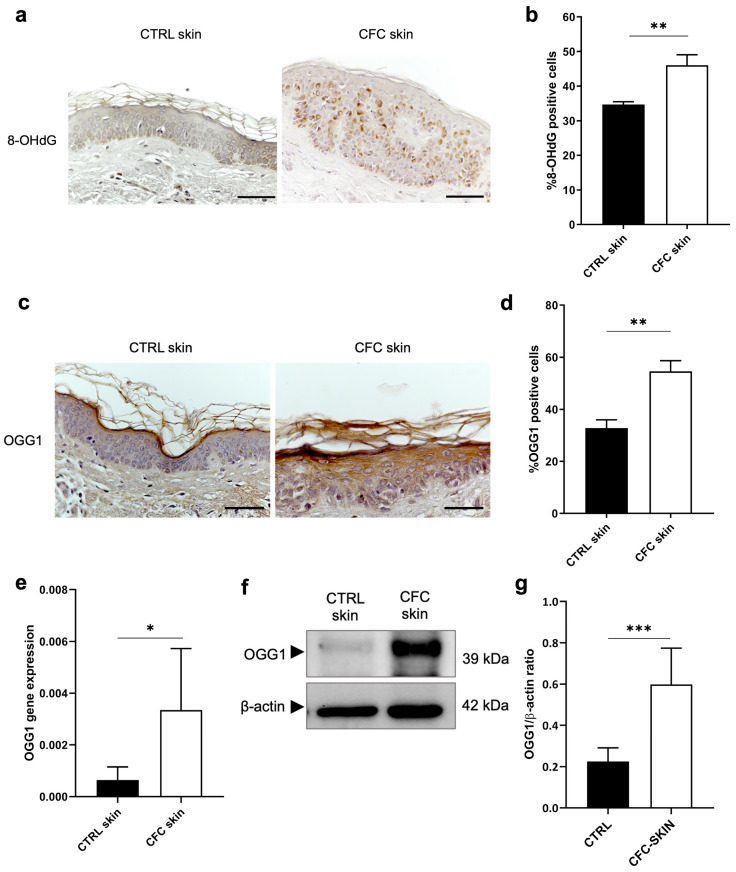
Evaluation of OGG1 expression as marker of DNA damage. (**a**) Representative IHC and (**b**) % of 8-OHdG positive cells of cells counted in four random areas of CTRL (*n* = 7) and CFC (*n* = 10) samples expressed as means ± SEM. (**c**) *OGG1* gene expression expressed as means ± SEM of CTRL (*n* = 12) and CFC (*n* = 21) skin samples. (**d**) Representative IHC expression and (**e**) % of OGG1 positive cells counted in four random areas of CTRL (*n* = 7) and CFC (*n* = 10) samples expressed as means ± SEM. (**f**) Representative Western blot and densitometric analysis (**g**) of OGG1 performed on CTRL (*n* = 6) and CFC (*n* = 10) samples expressed as means ± SEM. CTRL, control; CFC, cutaneous field cancerization; OGG1, 8-oxoguanine DNA glycosylase; 8-OHdG, 8-hydroxy-2-deoxyguanosine. Scale bar: 10 μm. * *p* < 0.05, ** *p* < 0.01, *** *p* < 0.001.

**Table 1 ijms-25-05775-t001:** Clinical characteristics CTRL subjects.

	Age	Sex	Anatomical Site
Ctrl_1	51	M	Trunk
Ctrl_2	41	M	Trunk
Ctrl_3	79	M	Trunk
Ctrl_4	67	M	Trunk
Ctrl_5	88	M	Trunk
Ctrl_6	86	M	Lower limb
Ctrl_7	84	F	Scalp
Ctrl_8	61	F	Head
Ctrl_9	39	F	Scalp
Ctrl_10	79	M	Face
Ctrl_11	62	M	Lower limb
Ctrl_12	47	F	Scalp
Ctrl_13	51	M	Scalp
Ctrl_14	76	F	Neck
Ctrl_15	75	M	Scalp
Ctrl_16	88	F	Trunk
Ctrl_17	90	M	Trunk
Ctrl_18	67	M	Upper limb
Ctrl_19	71	F	Scalp
Ctrl_20	65	M	Face
Ctrl_21	79	F	Scalp
Ctrl_22	85	F	Neck
Ctrl_23	89	M	Trunk
Ctrl_24	86	F	Scalp
Ctrl_25	75	M	Lower limb

CTRL, control; F, female; M, male.

**Table 2 ijms-25-05775-t002:** Clinical characteristics of CFC patients.

Patient	Age	Sex	Anatomical Site	Skin Cancer
CFC_1	83	M	Head	BCC
CFC_2	83	M	Trunk	BCC
CFC_3	82	M	Face	BCC
CFC_4	87	M	Face	BCC
CFC_5	91	M	Head	Keratoacanthomas
CFC_6	78	F	Face	BCC
CFC_7	84	M	Trunk	BCC
CFC_8	80	M	Trunk	BCC
CFC_9	83	M	Trunk	BCC
CFC_10	75	M	Head	SCC
CFC_11	82	M	Face	BCC
CFC_12	87	M	Neck	BCC
CFC_13	91	M	Head	Keratoacanthomas
CFC_14	78	F	Face	BCC
CFC_15	87	F	Face	BCC
CFC_16	67	M	Lower limb	BCC
CFC_17	86	M	Lower limb	Melanoma
CFC_18	87	M	Face	SCC
CFC_19	90	F	Face	BCC
CFC_20	72	M	Trunk	AK
CFC_21	83	M	Trunk	Epithelioma
CFC_22	85	M	Upper limb	BCC
CFC_23	83	M	Head	BCC
CFC_24	86	M	Scalp	KA
CFC_25	83	M	Head	Epithelioma
CFC_26	75	M	Face	BCC
CFC_27	77	F	Trunk	BCC
CFC_28	51	M	Trunk	BCC
CFC_29	74	M	Trunk	Melanoacanthoma
CFC_30	63	M	Trunk	BCC
CFC_31	89	M	Lower limb	SCC
CFC_32	86	M	Head	AK
CFC_33	67	M	Trunk	BCC
CFC_34	54	M	Trunk	Displastic nevi
CFC_35	78	F	Trunk	BCC
CFC_36	82	M	Upper limb	BCC
CFC_37	66	F	Trunk	BCC
CFC_38	64	F	Face	BCC
CFC_39	83	M	Head	BCC
CFC_40	83	M	Head	BCC
CFC_41	74	M	Lower limb	Melanoma

CFC, cutaneous field cancerization; BCC, basal cell carcinoma; SCC, squamous cells carcinoma; AK, actinic keratosis; M, male; F, female.

## Data Availability

The data presented on this study are available on request from the corresponding author.

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
