# Peer review of "Ex Vivo Analysis of Cell Differentiation, Oxidative Stress, Inflammation, and DNA Damage on Cutaneous Field Cancerization"

_ijms, 2024, doi:10.3390/ijms25115775_

Round 1

Reviewer 1 Report

Comments and Suggestions for Authors

Cutaneous field cancerization (CFC) is a concept that encompasses apparently normal tissue surrounding neoplastic lesions, primarily influenced by chronic exposure to ultraviolet radiation. Understanding CFC is crucial due to its association with heightened risks of precancerous and cancerous skin lesions. Overall, the authors' findings generally corroborate their conclusion. However, addressing the following points could enhance the manuscript's quality.

1.       While CFC has been utilized to characterize actinic dysplasia and keratinocyte carcinomas (KCs), a definitive and succinct definition remains elusive. Is there an effective grading system for CFC?

2.       Studies have indicated that p53-clonal fields play a pivotal role in CFC formation, with TP53 and NOTCH mutations recognized as early drivers in its progression. The authors should elaborate on how their current findings contribute to existing knowledge.

3.       It's important to explore the relationship between the authors' findings and actinic keratosis (AK) tissue associated with squamous cell carcinoma (SCC). For instance, understanding how CFC exacerbates AK leading to SCC.

4.       Clarification is needed on the significance of dermal staining of p16. If this is true staining, what implications does it carry?

5.       The authors need to address the impact of UV-induced changes in the dermal microenvironment, such as damage to the extracellular matrix (ECM) and inflammation, on CFC.

Author Response

  1. While CFC has been utilized to characterize actinic dysplasia and keratinocyte carcinomas (KCs), a definitive and succinct definition remains elusive. Is there an effective grading system for CFC?

To the best of our knowledge there is no grading system for CFC. However, to try to answer this question we have integrated the Introduction and Discussion section with some sentences and with related references. In detail, we referred to the 8th AJCC, in which are distinguished SCCs arising on photo exposed areas from those arising on non-photo exposed areas. Furthermore, we referred to the new nomenclature proposed by Conforti et al (JEADV 2019), based on the onset of SCC from the cancerization field.

  1. Studies have indicated that p53-clonal fields play a pivotal role in CFC formation, with TP53 and NOTCH mutations recognized as early drivers in its progression. The authors should elaborate on how their current findings contribute to existing knowledge.

As suggested by the reviewer, we explained in the discussion part how our findings could contribute to fill the gap of the current knowledge (lines 217-222).

  1. It's important to explore the relationship between the authors' findings and actinic keratosis (AK) tissue associated with squamous cell carcinoma (SCC). For instance, understanding how CFC exacerbates AK leading to SCC.

As already explained in the answer to question 1, the role of CFC in the onset of AK and in the evolution into SCC is demonstrated by the identification of two specific categories, i.e. the SCC related, and the SCC not related to the photo exposure.

  1. Clarification is needed on the significance of dermal staining of p16. If this is true staining, what implications does it carry?

We confirm that the dermal p16 stain is the actual expression of the protein on fibroblasts. Since we decided to focus our attention mainly on the epidermis, we do not have any data available that allow us to speculate about the possible implications of p16 on dermal fibroblasts. However, considering the lack of knowledge about this topic, we could explore this interesting finding on our next studies.

  1. The authors need to address the impact of UV-induced changes in the dermal microenvironment, such as damage to the extracellular matrix (ECM) and inflammation, on CFC.

As explained in the answer to question 4, in this paper we mainly focused on epidermal changes, and we are not able to correlate UV-induced changes in the dermal microenvironment. However, we could study in deep this interesting aspect of CFC development on next studies.  We added a short sentence on the Conclusion section about reviewer’s suggestion.

Reviewer 2 Report

Comments and Suggestions for Authors

SUMMARY

In this interesting paper the authors investigated the features of cutaneous field cancerization comparing biopsies from patients with known skin lesions with skin biopsies from healthy donors. They performed multiple molecular analysis including IHC, immunofluorescence, protein and RNA expression and observed different levels of expression of tumor suppressor genes and pro-inflammatory molecules in the two groups.

MINOR

The abstract state: “We biopsied perilesional skin from 41 non-melanoma skin cancer patients”…however there were 2 melanoma patients in the cohort, please explain.

Line 33  “severe”…I think the authors mean severely or several.

Line 34 is CFC characterized also by higher risk of melanoma?

Line 47 “The most mutated genes on both normal sun-exposed skin and AKs are”…I think authors mean “the most frequently mutated genes”; the same for line 211.

Line 72 explain CTRL

Line 192 please clarify the meaning of “subclinical skin area”

Line 206-210, please reformulate this sentence, it is very difficult to understand

Line 224 please clarify what do you mean by interesting

Line 284 “all donors signed the informed consent approved by the competent”…who is the competent? I think authors means the etic board?

Please add references in the Methods. Many procedure are reported as “as described before” but there is no reference.

The procedure for RNA extraction is not reported.

I believe the conclusions should be after the discussion.

MAJOR

The discussion is too long and should be re-organized. The first part reports exactly the same concepts already presented in the introduction. The second part describe the results in detail. I think the authors should avoid repetitions and use this section to discuss the implications of their findings, why are they important? how do they change our understanding of CFC? How can they be used to improve patients care? Why do you think CDK inhibitors are increased? Etc..

Another limitation is that there is no comparison between the characteristics of patients and healthy controls. Was the difference in median age significant? If yes might have impacted the study results. Were the healthy control and the patients matched? Please clarify.

Comments on the Quality of English Language

Please check the quality of the English throughout the whole manuscript. Overall sentences are too long and sometimes difficult to understand.

Author Response

MINOR

The abstract state: “We biopsied perilesional skin from 41 non-melanoma skin cancer patients”… however there were 2 melanoma patients in the cohort, please explain.

We corrected the abstract as suggested by the reviewer.

Line 33 “severe”…I think the authors mean severely or several.

We delete “severe” because was not pertinent.

Line 34 is CFC characterized also by higher risk of melanoma?

Line 47 “The most mutated genes on both normal sun-exposed skin and AKs are”…I think

authors mean “the most frequently mutated genes”; the same for line 211.

We corrected as indicated by the reviewer.

Line 72 explain CTRL

We explained within the text.

Line 192 please clarify the meaning of “subclinical skin area”

We removed “subclinical” and we reformulated the sentence.

Line 206-210, please reformulate this sentence, it is very difficult to understand

We reformulated the sentence according to reviewer’s suggestion.

Line 224 please clarify what do you mean by interesting

We deleted “interesting” because was not pertinent.

Line 284 “all donors signed the informed consent approved by the competent”…who is the

competent? I think authors means the etic board?

We better specified as suggested by the reviewer.

Please add references in the Methods. Many procedure are reported as “as described before”

but there is no reference.

We specified in the text that “as described before” refers to paragraph 4.2 of results where we described how epidermis lysis was performed depending on the assay.

The procedure for RNA extraction is not reported.

We added in paragraph 4.5.

I believe the conclusions should be after the discussion.

We followed IJMS template where conclusions are at the end of the paper (after materials and methods section).

MAJOR

The discussion is too long and should be re-organized. The first part reports exactly the same concepts already presented in the introduction. The second part describe the results in detail. I think the authors should avoid repetitions and use this section to discuss the implications of their findings, why are they important? how do they change our understanding of CFC? How can they be used to improve patients care? Why do you think CDK inhibitors are increased?

Etc..

The Discussion has been extensively revised, based on the suggestions.

Another limitation is that there is no comparison between the characteristics of patients and healthy controls. Was the difference in median age significant? If yes might have impacted the study results. Were the healthy control and the patients matched? Please clarify.

Patients and healthy controls weren’t matched; however, differences in terms of gender composition and median age were not statistically significant. This point was specified in the appropriate section of Materials and Methods (4.1) and also in the Conclusion, as a possible limitation for this study.

Comments on the Quality of English Language

Please check the quality of the English throughout the whole manuscript. Overall sentences are too long and sometimes difficult to understand.

The English language has been revised, with the help of a native translator.

Round 2

Reviewer 1 Report

Comments and Suggestions for Authors

Thank for your response.

Author Response

Thank you very much for your time and efforts in revising our manuscript.

Reviewer 2 Report

Comments and Suggestions for Authors

My prior comments have all been addressed. I have no further comments for the authors

Author Response

Thank you very much for the time and efforts in revising our manuscript.